# Disrupting Insulin and IGF Receptor Function in Cancer

**DOI:** 10.3390/ijms22020555

**Published:** 2021-01-08

**Authors:** Jingran Cao, Douglas Yee

**Affiliations:** 1Department of Pharmacology, University of Minnesota, Minneapolis, MN 55455, USA; cao00072@umn.edu; 2Masonic Cancer Center, University of Minnesota, Minneapolis, MN 55455, USA; 3Department of Medicine, University of Minnesota, Minneapolis, MN 55455, USA

**Keywords:** insulin-like growth factors, type I IGF receptor, insulin receptor

## Abstract

The insulin and insulin-like growth factor (IGF) system plays an important role in regulating normal cell proliferation and survival. However, the IGF system is also implicated in many malignancies, including breast cancer. Preclinical studies indicate several IGF blocking approaches, such as monoclonal antibodies and tyrosine kinase inhibitors, have promising therapeutic potential for treating diseases. Uniformly, phase III clinical trials have not shown the benefit of blocking IGF signaling compared to standard of care arms. Clinical and laboratory data argue that targeting Type I IGF receptor (IGF1R) alone may be insufficient to disrupt this pathway as the insulin receptor (IR) may also be a relevant cancer target. Here, we review the well-studied role of the IGF system in regulating malignancies, the limitations on the current strategies of blocking the IGF system in cancer, and the potential future directions for targeting the IGF system.

## 1. Introduction

The insulin and insulin-like growth factor (IGF) system is critical for normal growth and development by regulating cell growth, differentiation, survival, and metabolism [1,2]. The IGF system is composed of three ligands: IGF-1, IGF-2, and insulin; six high affinity ligand-binding proteins (IGFBPs) and several receptors, including the type I IGF receptor (IGF1R), type II IGF receptor (IGF2R), insulin receptor (IR), and hybrid receptors composed of one chain of the IGF1R and one chain of the IR [3].

Among the above three receptors in the IGF system, IGF1R and IR belong to the receptor tyrosine kinases (RTKs) family, and IGF2R does not possess this biochemical activity. Many growth factors bind and activate RTKs by inducing the receptor dimer formation [4]. However, cell surface IGF1R or IR exists as homodimer or heterodimer without a requirement for ligand binding. Each dimer is composed of two disulfide-linked polypeptide chains, also called a “half receptor”. Each half-receptor consists of an extracellular α-chain containing the ligand-binding domain and an intracellular β-chain with tyrosine kinase activities [5,6]. Previous studies have revealed that IGF1R and IR share a high similarity in both ligand-binding domains and kinase domains [7]. IGF1R or IR ectodomain monomer exists as an inverted “V”-shape; one leg of “V” consists of the first leucine-rich repeat domain (L1), the cysteine-rich region (CR), and the second leucine-rich repeat domain (L2); the other leg of “V” is composed of three fibronectin type III domains (FnIII-1, FnIII-2, and FnIII-3) and an insert domain (ID), which is located within the CC′ loop of FnIII-2 and contains the C-terminal region of α-chain (αCT) [8,9].

For the inactive receptor dimer, the connection of the L1 domain of one monomer and the αCT segment of the other monomer is critical for forming a ligand-binding site because of their proximity and corresponding electron density [7]. IGF1R and IR differ slightly in the L1 and CR domains. As a result, different ligand-binding sites formed by receptors lead to differential ligand-binding preferences [7]. IGF-1 has a higher affinity to IGF1R and hybrid IGF1R/IR than to IR, insulin has a greater affinity for IR than IGF1R and hybrid receptors, while IGF-2 binds to IGF1R, IR-isoform A (IR-A) or their hybrid receptors with high affinity compared with IR-isoform B (IR-B) [10,11]. Unlike IGF1R and IR, IGF2R lost the intracellular domain. Instead of transducing signals, IGF2R mainly works by removing IGF-2 by receptor-mediated endocytosis and lysosome degradation [12].

Once the growth factor ligand binds to the extracellular domain of the receptors, the intracellular tyrosine kinase domain (TKD) of one β-subunit phosphorylates its apposing strand resulting in receptor autophosphorylation. Phosphorylated tyrosine residues recruit adaptor proteins, including insulin receptor substrate (IRS)-1, IRS-2 and SHC. Phosphorylation and activation of adaptor proteins induce the activation of downstream signaling such as PI3K/Akt and MAPK pathways, therefore regulating cell growth, survival, apoptosis, metabolism, and mediating normal physiological processes at both cellular level and systemic level [13].

## 2. Rationale for Targeting the IGF System in Diseases

As mentioned, IGF1R and IR are widely expressed in normal human tissue and play important roles in supporting the physiological function of the human body. Epidemiological studies also suggest that the IGF system is extensively associated with the development and progression of several diseases such as cancer [14,15]. Components of the IGF system are commonly expressed in human cancers such as prostate cancer, breast cancer, and lung cancer [16,17,18]. Dysregulation of the IGF system contributes to the progression of multiple chronic liver diseases, such as nonalcoholic fatty liver disease (NAFLD), which may gradually promote hepatocarcinogenesis [19]. Several studies suggest that the expression of IGF1R is negatively correlated with the disease-free survival in non-small cell lung cancer (NSCLC) [20,21]. In addition, the IGF system contributes to the progression of breast cancer, colon cancer, and prostate cancer [22]. Epidemiologic studies have shown that a high circulating concentration of IGF-1 is associated with an increased risk of prostate cancer and breast cancer [23,24,25]. A high level of circulating insulin is also related to higher risk and increased mortality of colorectal and pancreatic cancer [26,27].

The IGF system contributes to not only malignant diseases, but also to autoimmune diseases such as thyroid eye disease (TED) or thyroid-associated ophthalmopathy (TAO) [28]. It is proposed that activation of the thyroid-stimulating hormone receptor (TSHR) by activating autoantibodies leads to excessive production of thyroid hormones and eventually causes TED [29,30]. Although the pathogenesis of TED is still incompletely understood, it is increasingly clear that IGF1R and IGF-1 are involved in the development and progression of TED. IGF1R is overexpressed in TED, and anti-IGF1R autoantibodies have been detected in TED patients [31,32,33]. A functional complex formed by TSHR and IGF1R has been found in orbital fibroblast of TED patients [34]. Additionally, IGF-1 enhances the function of thyroid-stimulating hormones (TSH) [35]. The above evidence suggests that activation of IGF1R and its crosstalk with TSHR are implicated in the pathogenesis of TED.

Preclinical studies have well-characterized the role of the IGF system in promoting diseases, especially in cancer. IGF-1 and IGF-2 are mainly produced in the liver, although some of them can also be synthesized from other tissue such as kidney, brain, or neoplastic tissue, regulating normal physiological processes or tumor growth by autocrine, paracrine, and endocrine manners [36,37,38]. IGF-1 promoted proliferation and survival in triple-negative breast cancer (TNBC) cell lines by activating IGF1R signaling, and the MDA-MB-231 cell line had the most significant response. Knockdown of the IGF1R in TNBC cells lower the growth-promoting effects of IGF-1 [39]. Mice bearing IGF-1- and IGF-2-overexpressing MCF-7L cells had earlier onset of tumorigenesis and increased tumor growth rate compared with mice bearing control MCF-7L cells, which is probably due to increased amino acid synthesis under the regulation of IGF signaling [40]. In liver-specific IGF-1 gene-deleted (LID) mouse model, incidence of mammary tumors was lower and tumor formation was delayed compared with control mice [41]. IGF1R also plays an important role in promoting cancer metastasis as reviewed [42]. In addition to cancer growth and metastasis, increasing evidence suggests that The IGF system also contributes to cancer therapeutic resistance. Increased IGF1R signaling renders NSCLC cells more tolerant to osimertinib, an EGFR-tyrosine kinase inhibitor [43,44]. Dysregulation of the IGF system is associated with tamoxifen resistance in breast cancer [45,46,47].

Recently, IR has also been considered as a cancer target. IR exists in two isoforms, IR-A and IR-B. IR-A differs from IR-B by the exclusion of exon 11, which is thought to contribute to ligand preference and therefore differential receptor activation. IR-A is associated with mitogenic pathways and mainly expressed in cancer and fetal cells, while IR-B mainly regulates glucose homeostasis and is predominantly expressed in metabolic tissue [48]. Our data suggest that loss of IGF-1R expression renders endocrine-resistant breast cancer cells more sensitive to insulin, probably due to increased insulin binding sites [45], which activates the overlapping pro-tumorigenic pathways to IGF-1R. Pre-clinical studies have shown that IR can drive and accelerate breast tumor progression independently of IGF-1R in the hyperinsulinemia mouse model [49,50]. Recent studies have shown that IR is upregulated with a significantly high IR-A/IR-B ratio in endocrine-resistant breast cancer cells [51]. Our data have shown that disruption of IR by either shRNA or blocking peptide inhibits the growth in endocrine-resistant breast cancer cells [52]. Based on this work, specifically inhibiting IR-A and leaving IR-B undisrupted could be an ideal therapeutic strategy, since disruption of IR-B leads to metabolic dysfunction.

Both epidemiological and preclinical evidence suggests the components of the IGF system are important targets for therapeutic development. Here, we review the current anti-IGF strategies, and we mainly focus on the approaches targeting IGF1R and IR.

## 3. Methods of Targeting IGF1R and IR in Diseases

In theory, there are several ways this receptor system could be disrupted. Disruption of ligand–receptor interactions would prevent downstream activation of signaling. Alternatively, inhibition of the tyrosine kinase domain of the receptors would also block signal transduction. Finally, neutralization of ligand activation could also disrupt receptor activation. All of these strategies have been tested.

### 3.1. Monoclonal Antibodies

The monoclonal antibodies (mAbs) have been widely studied and used in cancer treatment. Most mAbs function by directly binding the receptor. Some will prevent the binding of growth factor ligands to their receptors, therefore inhibiting the activation of downstream signaling and the dysregulation of cellular activities. Others, such as trastuzumab, the first targeted-therapy approved by FDA for breast cancer, were designed as recombinant monoclonal antibodies against HER2 [53]. HER2 does not have a ligand and is commonly expressed at a low level in normal epithelial cells, but highly expressed in some breast cancer cells. Currently, several mechanisms of action have been proposed for trastuzumab, including the induction of HER2 degradation, induction of antibody-dependent cellular cytotoxicity, and the inhibition of intracellular signaling pathways [53]. However, the exact mechanism is still unclear. Even though the precise mechanism of action is not understood and may have multiple mechanisms, clinical trials showed that trastuzumab combined with chemotherapy significantly improves treatment outcomes for patients with HER2-positive breast cancer [54,55].

Both IGF1R and HER2 are receptor tyrosine kinases. Similar to trastuzumab, many mAbs against IGF1R were developed (Figure 1). Preclinical studies have shown that cixutumumab (IMC-A12), a mAb against IGF1R, blocked ligand binding and intracellular signaling activation in MCF-7L cells. Cixutumumab also had strong anti-tumor activities in xenograft tumor mouse models [56]. These promising results resulted in cixutumumab testing in clinical trials. However, the results of current clinical studies were disappointing. In a small phase II randomized trial of 93 patients with hormone-receptor-positive (HR+) breast cancer, cixutumumab alone, or in combination with antiestrogen therapy, did not show significant clinical benefits [51]. It is noted that cixutumumab induced several side effects such as hyperinsulinemia [57], which is probably due to the disruption of the GH/IGF negative feedback loop by IGF1R antibody [58,59]. Cixutumumab was also evaluated in another Phase II trial in patients with advanced NSCLC, no additional benefit was observed when cixutumumab was added to traditional chemotherapy [60].

Ganitumab (AMG-479), another anti-IGF1R mAb, blocked IGF1R activation and inhibited tumor growth in pancreatic tumor xenografts [61]. Ganitumab was then evaluated in a clinical trial of 125 patients with metastatic pancreatic cancer. In this study, ganitumab combined with gemcitabine showed benefits in 6-month overall survival [62]. However, in a phase II clinical trial of patients with advanced HR+ breast cancer, ganitumab did not show clinical benefits in this group of patients when added to endocrine therapy [63]. Previous data suggest that the failure of anti-IGF1R mAb is partly due to the loss of IGF1R and the increase of IR expression in endocrine-resistant breast cancer cells [45,64]. Therefore, we need to consider the timing of adding anti-IGF1R mAb to current endocrine therapy and the incorporation of predictive biomarkers when conducting clinical trials to evaluate the efficacy of anti-IGF1R mAbs as breast cancer therapeutics. Currently, results from clinical trials testing ganitumab, including a phase I/II trial in patients with rhabdomyosarcoma in combination with Src family kinase inhibitor dasatinib (NCT03041701); a phase II trial evaluating the efficacy of ganitumab combined with CDK4/6 inhibitor palbociclib in patients with Ewing sarcoma (NCT04129151) are pending.

In order to address hyperglycemia caused by IGF1R antibody, ganitumab combined with metformin and paclitaxel was evaluated in an I-SPY trial in breast cancer patients (NCT01042379). The initial reports of this trial showed no benefit of addition of ganitumab and metformin to conventional chemotherapy [65].

Although anti-IGF1R mAbs have not been approved as cancer therapeutics, teprotumumab, a fully human mAb against IGF1R, has recently been approved by the FDA for treating TED. As we mentioned early, IGF1R and IGF-1 play important roles in mediating the pathogenesis of TED, and several clinical studies indicate that teprotumumab significantly reduces proptosis in patients with TED [66,67,68]. The approval of teprotumumab marks an important breakthrough in the development of IGF1R antibodies.

### 3.2. Tyrosine Kinase Inhibitors

Several small-molecule tyrosine kinase inhibitors (TKIs) have been developed for targeting the IGF1R. Linsitinib (OSI-906) is an ATP-competitive TKI against both IGF1R and IR [69]. Preclinical studies show that linsitinib blocks the autophosphorylation of IGF1R/IR and exert promising anti-tumor activity [69]. However, a clinical trial evaluating the efficacy of linsitinib in patients with metastatic breast cancer was terminated at phase II due to severe toxicities such as hyperglycemia (NCT01205685). The side effect caused by linsitinib is mainly because of the disruption of IR, which plays an important role in maintaining glucose homeostasis [70]. BMS-754807, another ATP-competitive TKI of IGF1R and IR, shows growth inhibitory effects in pancreatic cancer cell lines [71,72]. However, a phase II clinical trial evaluating the effects of BMS-754807 alone or in combination with letrozole in breast cancer was terminated earlier than anticipated (NCT01225172). Results of this clinical trial have been reported recently, several side effects such as impaired glucose tolerance, fatigue, and nausea are seen in almost half of the patients.

Both linsitinib and BMS-754807 are dual IGF1R/IR inhibitors. Inhibition of IR function can lead to unwanted side effects, such as dysregulated glucose homeostasis. Given the IGF1R and IR share a 100% similarity in their ATP-binding site, other approaches have been developed to selectively inhibit IGF1R instead of directly targeting the ATP-binding site. Picropodophyllin (PPP), a member of cyclolignans, has been shown to selectively inhibit IGF1R function without interfering with IR activity [73]. Previous studies show that PPP inhibits tumor cell growth, and induces apoptosis and cell cycle arrest in multiple tumor cell lines and xenograft tumor models [74,75,76]. Several mechanisms have been proposed for the growth inhibitory effects of PPP, including the induction of IGF1R degradation and inhibition of IGF1R autophosphorylation [73,74,77,78], although the exact mechanism remains to be determined. AXL 1717, an orally active PPP, is being evaluated in an early phase I trial of patients with astrocytoma. Early clinical data suggest PPP is well-tolerated and has potential anti-tumor activity [79].

Allosteric tyrosine kinase inhibitors are also being developed to target IGF1R with specificity [80]. Heinrich et al. developed a series of compounds that function as allosteric inhibitors against IGF1R [81]. Crystallographic studies reveal that a representative compound 10, instead of directly binding to the ATP-binding site, binds to an adjacent pocket next to the activation loop of the tyrosine kinase domain. Moreover, biological studies indicate that another potent compound 11 (IC50 = 0.2 mM) does not disrupt IR signaling when concentration is 30 mM [81]. However, these compounds have not been evaluated in clinical trials yet. Another example of allosteric inhibitors is NT157 described by NovoTyr [82]. Binding of NT157 to IGF1R induces a conformational change of IGF1R. This allosteric regulation results in the detachment of IRS1/2 from IGF1R and inhibitory Ser-phosphorylation of IRS1/2, which eventually leads to the downregulation of IRS1/2 and inhibition of IGF1R signaling [82]. Preclinical studies indicate that NT157 inhibits growth in osteosarcoma, breast cancer, and myeloproliferative neoplasms [83,84,85]. However, no clinical data are currently available for NT157.

### 3.3. Peptide Inhibitors

Several small peptide inhibitors against IGF1R or IR have been developed and evaluated in both preclinical and clinical studies. S961, a small peptide IR antagonist developed by Novo Nordisk [86], has been used in studying the IR-associated disease such as hyperinsulinemia and breast cancer [87,88]. Previous work shows S961 inhibits insulin-stimulated growth and cell cycle progression in breast cancer cell lines [52]. While S961 was originally reported as an IR antagonist, researchers found that S961 also exhibits agonist effect at low concentration range [89]. Currently, the exact mechanism of S961 action is still not clear. Knudsen et al. proposed that a single S961 peptide can bind and activate the IR dimer. However, another S961 peptide may also be able to bind to the same IR dimer simultaneously, the conformational change induced by the second ligand binding converts activated IR into inactivated form, therefore exhibiting antagonistic effects [89]. Although S961 shows promising anti-tumor effects in cell-based assays, severe side effects such as hyperglycemia and hyperinsulinemia are seen in the mouse model of breast cancer [87].

As mentioned earlier, the side effects caused by dual IGF1R/IR inhibitors observed in clinical trials are probably due to the disruption of IR function. IR has two isoforms, IR-A and IR-B. IR-A differs from IR-B by the exclusion of exon 11. IR-A is more related to mitogenic signaling, while IR-B is more associated with metabolic signaling [48]. Given IGF1R and IR activated similar signaling pathways, previous studies propose that IR compensates for the loss of IGF1R and mediates cancer cell growth [45,52,90]. Therefore, co-targeting IR and IGF1R may be necessary to completely block the growth-stimulatory pathways. In addition, specifically targeting the mitogenic IR-A isoform, leaving the metabolic IR-B isoform undisrupted is needed to prevent glucose dysregulation. This proposed mechanism drives the development of the IR inhibitor with isoform specificity.

Recently, we described a small peptide Gp2 with inhibitory effects against IR without affecting IGF1R [91]. Three Gp2 variants bind to cell surface IR with low nanomolar affinity, while showing minimal binding to IGF1R, suggesting promising specificity to IR over IGF1R. Cell-based studies show that Gp2 variants inhibit insulin-mediated signaling activation and cell growth in several breast cancer cell lines, indicating Gp2 variants are able to block the function of IR. However, the exact mechanism of Gp2 action is still unknown and the specificity against specific IR isoform needs to be further studied.

### 3.4. Ligand Neutralization

Since ligand binding to IGF1R or IR is required to initiate signaling, neutralizing IGF ligands is another strategy to block receptor function. In 2011, Gao et al. reported that MEDI-573 (dusigitumab), a fully human antibody, binds to IGF-1 and IGF-2 with high affinity, therefore disrupting IGF binding to the receptors [92]. In this work, they show the binding of MEDI-573 to the ligands is able to prevent IGF-1-stimulated IGF1R activation and IGF-2-stimulated-IR-A activation. Moreover, cell-based assays and the xenograft mouse model indicate that MEDI-573 disrupt the growth of embryonic cell lines overexpressing IGF1R and IGF-1/2 [92]. These promising results enabled MEDI-573 to be evaluated in clinical trials. Early clinical data indicate MEDI-573 has a favorable safety profile without dose-limiting toxicities [93]. However, in the phase Ib/II trial evaluating the efficacy of MEDI-573 combined with aromatase inhibitors (AI) in patients with metastatic HR+ breast cancer, MEDI-573 did not show additional benefits over AI alone in progression-free survival (NCT01446159). In another phase Ib/II trial (NCT01498952), MEDI-573 was evaluated combined with sorafenib, a tyrosine kinase inhibitor of VEGF, in patients with unresectable or metastatic hepatocellular carcinoma (HCC). However, progression-free survival was not evaluated because the sponsor did not launch the phase II trial.

Another ligand neutralizing antibody developed by Boehringer Ingelheim, BI 836845 (xentuzumab), is also being evaluated in both preclinical and clinical studies. Similar to MEDI-573, binding of BI 836845 to IGF-1/2 results in ligand neutralization and inhibition of IGF1R and IR function. In addition, BI 836845 shows anti-tumor activities in several cancer cell lines and xenograft mouse models [94]. Unlike MEDI-573, the ligand binding site of BI 836845 is not clear. The pharmacodynamic effects of the two antibodies are different, which suggests that MEDI-573 and BI 836845 may have different therapeutic effects. It is noted that the above two ligand neutralizing antibodies do not induce severe metabolic disorders in preclinical animal models compared with other anti-IGF1R strategies [57]. BI 836845 has been tested in several clinical trials, including a phase Ib/II trial evaluating BI 836945 in combination with everolimus or exemestane in patients with estrogen receptor-positive breast cancer (NCT02123823), a phase Ib trial in patients with non-small cell lung cancer with afatinib (NCT02191891), and a phase Ib/II trial in patients with castrate-resistant prostate cancer with enzalutamide (NCT02204072). The results of these clinical trials have not been reported yet. Additionally, BI 836845 is being evaluated in two ongoing clinical trials, including a phase II trial in patients with HR+/HER2- metastatic breast cancer with everolimus and exemestane (NCT03659136), a phase Ib trial in patients with different types of solid tumors plus CDK4/6 inhibitor abemaciclib (NCT03099174). Early clinical results of NCT03099174 show that the combination of xentuzumab with abemaciclib has an acceptable safety profile [95].

### 3.5. Receptor Downregulators

Similar to the monoclonal antibodies (Table 1), PPP has been shown to downregulate receptor levels [78]. In addition to this compound, additional novel strategies can also be used to downregulate either IGF1R or IR. Nutlin-3 has been used to disrupt the binding of p53 and mouse double-minute 2 homolog (Mdm2). Mdm2 is an E3 ubiquitin ligase, promoting the ubiquitination of IGF1R and subsequent degradation. Melanoma cells with wild-type p53 showed downregulation of IGF1R, inhibited cell growth, and decreased IGF-stimulated migration after receiving Nutlin-3 treatment [96]. This work indicates Nutlin-3 could be a promising small molecule inhibitor against IGF1R, outlining a new approach to inducing IR degradation in cancer cells.

IR can also be downregulated using novel strategies. AKS-130, an insulin-Fc fusion protein developed by Akston Biosciences (Beverly, MA, USA), is being investigated as a long-acting insulin. Preclinical studies show that IR expression is downregulated in colon cancer cells after AKS-130 treatment, which is probably due to ligand-dependent endocytosis. In accordance, our data indicate that AKS-130 downregulates IR and partially blocks PI3K/Akt signaling in both MCF-7L and Tamoxifen-resistant (TamR) MCF-7L cells. Additionally, AKS-130 inhibits insulin-stimulated growth in TamR MCF-7L cells [97]. However, the in vivo effects of AKS-130 and the binding preference for different IR-isoform are still unknown.

### 3.6. Antisense Oligonucleotides

Antisense oligonucleotides (ASOs), one of the molecular agents for inhibiting the translation of IGF1R mRNA, are used in experimental studies unlike the inhibitors discussed above. Prostate cancer cell lines showed inhibited growth and increased apoptosis when transfected with ATL1101, a modified ASO against IGF1R. In the xenograft mouse model of prostate cancer, tumor growth rate was reduced after ATL1101 treatment [98]. Mice bearing C4HD tumor showed decreased tumor growth rate after receiving AS[S]ODNs, a phosphorothioate antisense oligodeoxynucleotide against IGF1R, via either intratumor injection or intravenous injection [99]. Although ASO showed anti-tumor activities in preclinical studies, the clinical application of ASO is still limited by specificity, efficiency, and stability. Therefore, no clinical trials evaluating IGF1R-targeted ASO has been launched.

## 4. Conclusions

IGF1R has been investigated as a therapeutic target in treating cancer for decades. Even preclinical studies show that targeting IGF1R inhibits cancer cell progression, and results of clinical trials evaluating anti-IGF1R strategies as cancer treatment have been disappointing. Many factors contribute to the failure of previous anti-IGF1R drugs. First is the identification of an “IGF driven” tumor subtype. Our preclinical work shows IGF1R is lost in endocrine-resistant breast cancer and this was confirmed in a clinical trial of endocrine-resistant tumors [51]. In the absence of the target in hormone resistant breast cancer, IGF1R mAb would be expected to be ineffective.

Second is the crosstalk and compensation between IGF1R and other growth-stimulating receptors, such as the insulin receptor. As mentioned above, IGF1R and IR are highly homologous, activating similar signaling pathways including PI3K/Akt and MAPK pathways. Therefore, IR can compensate for the loss of IGF1R and stimulate growth in endocrine-resistant breast cancer cells. Since IGF2 binds IR with high affinity, then IR inhibition would seem necessary. Preclinical studies showed inhibition of IR, either by shRNA or mAb disrupted endocrine-resistant breast cancer cell growth. However, IR inhibitor also results in several side effects, such as hyperinsulinemia caused by IGF1R mAb, which targets both IGF1R and IR without selectivity.

The structure of IR and IGF1R suggest that more than one targeted agent could be effective to disrupt signaling. To date, the IGF1R mAbs and TKIs were disappointing for several reasons, including their toxicities associated with disrupting glucose homeostasis. However, in the conduct of these trials we have learned that IR may also be a target. More promising is the potential to develop an inhibitor of IR that is more specific for cancer cells. As discussed previously, targeting IR-A is a promising strategy in developing novel anti-cancer therapeutics without inducing severe side effects. However, additional development will be required to demonstrate a molecule with specific IR-A isoform inhibition. If this could be achieved, then a more robust approach to inhibition of IGF and insulin signaling could be tested in cancer.

## Figures and Tables

**Figure 1 ijms-22-00555-f001:**
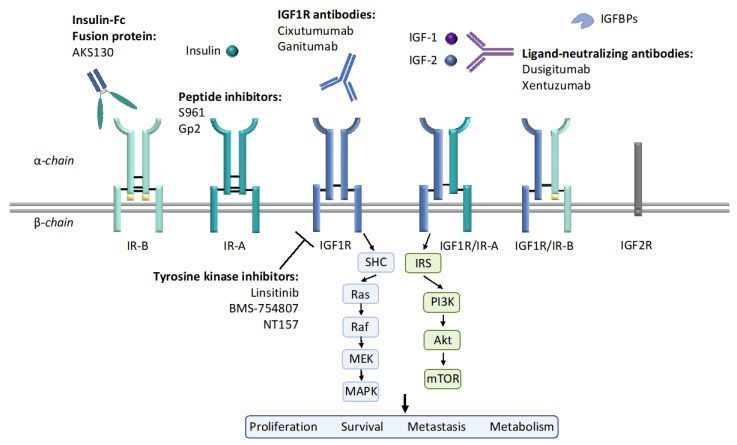
The insulin-like growth factor (IGF) system components and current anti-IGF1R/insulin receptor (IR) strategies. The IGF system is composed of three ligands: IGF-1, IGF-2, and insulin; six high affinity ligand-binding proteins (IGFBPs) and several receptors: IGF1R, IGF2R, IR, and hybrid receptors. IGF1R and IR have tetrameric structures, composed of either homodimer or heterodimer. Each dimer includes an extracellular α-chain and an intracellular β-chain, as indicated in this figure. IR-A differs from IR-B by the exclusion of the exon 11 encoded 12 amino acids at the C-terminal region of the α-chain. Upon ligand binding, downstream signaling such as PI3K/Akt and MAPK are activated, therefore regulating cell proliferation, survival, metastasis and metabolism. Current strategies for inhibiting IGF1R and IR include IGF1R antibody, IGF-1/2 neutralizing antibodies, tyrosine kinase inhibitors (TKIs), peptide inhibitors, etc.

**Table 1 ijms-22-00555-t001:** Strategies of targeting IGF1R and IR in diseases.

Drug Type	Compound	Preclinical Studies	Clinical Studies	Ongoing Clinical Trials
**Monoclonal antibodies**	Cixutumumab (IMC-A12)	[56]	[51,60]	
Ganitumab (AMG-479)	[61]	[62,63,65]	NCT03041701 NCT04129151 NCT01042379
Teprotumumab		[66,67,68]	
**Tyrosine kinase inhibitors**	Linsitinib (OSI-906)	[69]	NCT01205685	
BMS-754807	[71,72]	NCT01225172	
Picropodophyllin	[73,74,75,76]	[79]	NCT01721577
NT157	[82,83,84,85]		
**Peptide inhibitors**	S961	[86,87,88]		
Gp2	[91]		
**Ligand neutralization**	Dusigitumab (MEDI-573)	[92]	NCT01446159 NCT01498952	
Xentuzumab (BI 836845)	[94]	NCT02191891 NCT02123823 NCT02204072	NCT03659136 NCT03099174
**Receptor downregulators**	Nutlin-3	[96]		
AKS-130	[97]		
**Antisense oligonucleotides**	ATL1101	[98,99]

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
