# Peer review of "Disrupting Insulin and IGF Receptor Function in Cancer"

_ijms, 2021, doi:10.3390/ijms22020555_

Round 1

Reviewer 1 Report

The Authors provide a comprehensive overview of the current literature in the field of Insulin/IGF singling, with particular reference to the available pharmacological strategies aimed at normalizing this complex transduction pathway in cancer.

The manuscript is well organized and delivers the message in a straightforward manner. I would just add a table with the description of each blocking strategy described in the paragraphs, with particular reference to the ongoing clinical trials. This would help the reader to focus on the main output of the review. 

Author Response

Response to Reviewer 1 Comments

We appreciate the reviewer’s comments and the general favorable remarks. We have addressed the reviewers’ concerns as outlined below.

Reviewer:

The Authors provide a comprehensive overview of the current literature in the field of Insulin/IGF singling, with particular reference to the available pharmacological strategies aimed at normalizing this complex transduction pathway in cancer.

The manuscript is well organized and delivers the message in a straightforward manner. I would just add a table with the description of each blocking strategy described in the paragraphs, with particular reference to the ongoing clinical trials. This would help the reader to focus on the main output of the review.

Response:

Thank you for providing feedback. We have included a table summarizing clinical trials and their clinicaltrials.gov reference number. In the text of the manuscript, we have highlighted informative trial results.

Reviewer 2 Report

This is a well-done review of the role of the IGF system in regulating malignancies, the current strategies of blocking the IGF system in cancer, and the potential future direction of targeting the IGF system. The literature is summarized well. 

Comments:

1.        In the manuscript, the authors reviewed the rationale for targeting the IGF system in diseases and the methods of targeting IGF1R and IR in diseases. The abstract seems like describe the current situation, but not summarize the main points of the manuscript. It should be revised accordingly.

2.        Since there are several methods of targeting IGF1R and IR in diseases, it is highly recommend adding a table that shows all of them to make it easier to grasp.

3.        Please provide more detail information about the therapeutic agents targeting IGF system in clinical studies, if there is any available.

Author Response

Response to Reviewer 2 Comments

We appreciate the reviewer’s comments and the general favorable remarks. We have addressed the reviewer’s concerns as outlined below.

Reviewer 2:

This is a well-done review of the role of the IGF system in regulating malignancies, the current strategies of blocking the IGF system in cancer, and the potential future direction of targeting the IGF system. The literature is summarized well.

Comments:

Point 1: In the manuscript, the authors reviewed the rationale for targeting the IGF system in diseases and the methods of targeting IGF1R and IR in diseases. The abstract seems like describe the current situation, but not summarize the main points of the manuscript. It should be revised accordingly.

Response 1: Thanks for pointing it out. We have revised the abstract to summarize our main points.

Point 2: Since there are several methods of targeting IGF1R and IR in diseases, it is highly recommend adding a table that shows all of them to make it easier to grasp.

Response 2: As suggested by the reviewer, we have included a table summarizing blocking strategies and relevant clinical trials.

Point 3: Please provide more detail information about the therapeutic agents targeting IGF system in clinical studies, if there is any available.

Response 3: For each compound, we have summarized the information about its preclinical studies, completed clinical studies and ongoing clinical trials.

Reviewer 3 Report

In this Review, the authors provide a discussion of the latest developments in IGF-1 and insulin targeting and targeting of their receptor in various cancers. In so doing they also provide insight into what may be required in the future to make targeting of the insulin receptor feasible.

Overall,  I have no concerns nor do I think anything was left out of this manuscript. I do however, believe that the authors provided an excellent overview of the structures of the IGF1R and IR in lines 28-40 of the Introduction. This succinct description would be highly complemented by a graphic figure highlighting the structural features of the IGF1R, IRA and IRB.

Grammatical issues are present at the following lines:

36 - is consist should be consists of

48 - signaling should be signals; for should be by

51 - opposing should be apposing

56 - should be ...both by cellular...

68 -  , should be "and" (prostate cancer and breast cancer

79 - suggest should be suggests

197 - should be induces

201 - remains to be determined

218 - has should be have

292 - should be "downregulates" / should be "blocks"

298 - is should be "are used in experimental studies unilkethe inhibitors discussed above"

311 - clinical should be drugs

316 - early should be "above"

Author Response

Response to Reviewer 3 Comments

We appreciate the reviewer’s comments and the general favorable remarks. We have addressed the reviewer’s concerns as outlined below.

Reviewer 3:

In this Review, the authors provide a discussion of the latest developments in IGF-1 and insulin targeting and targeting of their receptor in various cancers. In so doing they also provide insight into what may be required in the future to make targeting of the insulin receptor feasible.

Overall, I have no concerns nor do I think anything was left out of this manuscript. I do however, believe that the authors provided an excellent overview of the structures of the IGF1R and IR in lines 28-40 of the Introduction. This succinct description would be highly complemented by a graphic figure highlighting the structural features of the IGF1R, IRA and IRB.

Response: 

1. Thanks for your time and feedback. We have substituted the old figure with a new figure to show the structural features of these receptors. For example, we show the alpha and beta chains as two separate peptide chains with covalent linkage instead of the previous single chain. We also show the 12 amino acids encoded by exon 11 to distinguish between IR-A and IR-B.

2. Thanks for pointing out grammatical issues. We have revised them accordingly.

Round 2

Reviewer 1 Report

The Authors have addressed my concerns.

The revised manuscript is overall improved and now suitable for publication, in my opinion.